# Association of Endogenous Testosterone with Physical Fitness Measures during Firefighter Occupational Health Evaluations

**DOI:** 10.3390/ijerph21030274

**Published:** 2024-02-27

**Authors:** Luiz Guilherme Grossi Porto, Edgard M. K. V. K. Soares, Sushant M. Ranadive, Adriana Lofrano-Porto, Denise L. Smith

**Affiliations:** 1Study Group on Exercise and Physical Activity Physiology and Epidemiology (GEAFS), Exercise Physiology Laboratory, Faculty of Physical Education, University of Brasilia—UnB, Brasilia 70910-900, Brazil; luizporto@unb.br (L.G.G.P.); esoares@skidmore.edu (E.M.K.V.K.S.); 2Department of Health and Human Physiological Sciences, Skidmore College, Saratoga Springs, NY 12866, USA; 3Department of Kinesiology, School of Public Health, University of Maryland, College Park, MD 20742, USA; ranadive@umd.edu; 4Adrenal and Gonadal Diseases Clinic, Section of Endocrinology and Metabolism of the University Hospital, University of Brasilia—UnB, Brasilia 70910-900, Brazil; adlofrano@unb.br

**Keywords:** public safety, workforce, cardiometabolic, cardiovascular, risk

## Abstract

Firefighting is a physically demanding profession associated with unacceptably high on-duty cardiovascular mortality. Low endogenous total testosterone (TT) is an emerging cardiometabolic (CM) risk factor in men, but limited data exists on its interactions with physical fitness (PF). Data from occupational health and fitness assessments of 301 male career firefighters (FFs) were analyzed. TT was categorized as low (<264 ng/dL), borderline (264–399 ng/dL), and reference (400–916 ng/dL). PF tests included cardiorespiratory fitness (submaximal treadmill), body fat percentage (BF%), push-ups, plank, and handgrip strength assessments. In the crude analyses, FFs in the low TT group had worse muscular and cardiorespiratory fitness measures compared to the referent group. However, after adjusting for age and BF%, none of the PF differences remained statistically significant. Similarly, the odds of less-fit FFs (PF performance below median values) having low TT were higher compared to the fitter ones only before adjusting for age and BF%. Therefore, in the final adjusted model, there was no significant association between TT and PF. Our data suggest that age and body fat confound the association between PF and TT. Low TT and poor PF are important components of FFs’ CM risk profile, and there is potential benefit to considering TT screening as part of a comprehensive occupational health program that manages performing medical evaluations and provides education and preventative programming.

## 1. Introduction

Firefighters (FFs) perform various duties, many of which involve strenuous physical work and emotional stress [1,2]. Although they vary worldwide, most often, FFs perform fire suppression, technical rescues, and emergency medical services. FFs’ routine tasks include stair climbing, forceable entry, crawling/searching, advancing hoselines and suppressing fire, and rescuing animals/persons. Importantly, these activities are done while wearing heavy protective equipment (~25 kg) and working in extreme environmental conditions with time urgency [1]. To perform these highly demanding activities, FFs need adequate physical fitness (PF), including both muscular and cardiorespiratory fitness (CRF) and proper body composition [1]. The US National Fire Protection Association (NFPA) recommends a minimum CRF of 12.0 metabolic equivalents (MET) for FFs to safely and efficiently perform their job-related tasks [3]. Also, this cut-off point has been widely used in firefighter fitness-related research [4,5,6]. Due to the hazards they face and the physiological strain of their work, firefighting is recognized as a hazardous profession with high on-duty mortality rates [7,8,9].

The US Fire Administration (USFA) reports that sudden cardiac events were responsible for almost 50% of on-duty deaths in the last 20 years among US FFs [9]. The odds of a cardiac event are highest following fire suppression activities [7,8]. While firefighting does not cause cardiac events, firefighting may trigger a cardiac event in individuals with underlying diseases [7,8]. A review of medical examiner records for duty-related cardiac fatalities of male FFs showed that almost 80% had evidence of coronary heart disease and increased heart size (cardiomegaly and/or left ventricular hypertrophy) at autopsy [8].

Furthermore, overweight/obese FFs and those with low or inadequate physical fitness (PF) have been shown to have worse job performance and undesirable cardiometabolic (CM) risk profiles [10,11]. Therefore, it is necessary to investigate further the relationship between cardiorespiratory and muscular fitness levels and the appropriate CM risk profiles required to safely perform the strenuous work of firefighting and decrease the risk of duty-related cardiac events [12,13,14,15].

In the context of complex CM risk assessments, endogenous serum testosterone (TT) levels have recently appeared as an emerging cardiovascular risk factor. In the general population, low TT has been associated with increased mortality [16,17]. In the first study to report on TT levels among US FFs, we found a proportion of 10.6% of low TT. FFs with lower testosterone were older and showed worse CM profiles than FFs with higher TT values [18]. Furthermore, we observed an association between borderline-low TT (264–319 ng/dL) and lower left ventricular wall thickness. In a more recent study among young/middle-aged career FFs, we observed a high proportion of fatty liver (FL; ≈40%), especially in those with low TT (78.1%). Having low TT also increased the odds of having a fatty liver and an unfavorable CM risk profile, even after adjusting for covariates [19]. However, despite the known physiological effects of TT on the cardiovascular and muscular systems [20], whether low TT levels influence PF remains unclear, even in the general population. We then hypothesized that low TT levels are associated with worse physical fitness measured in the context of occupational health assessment in firefighters. Therefore, the purpose of this study is to explore the relationship between TT with PF parameters among US career FFs.

## 2. Materials and Methods

### 2.1. Study Design and Participants

In this cross-sectional study, we analyzed records from an occupational health clinic that provides annual medical evaluations for FFs. Our sample included US male career FFs from a single fire department who have PF tests and TT assessment as part of their annual occupational health evaluation. This study is a novel arm of our research on FF health [18,19]. Since the researchers received retrospective de-identified data, the Skidmore College Institutional Review Board waived informed consent and approved the study protocol. This study was performed according to the Declaration of Helsinki.

This study population included all male FFs (n = 341) in a Florida fire department who were cleared for firefighting duty following their medical evaluation. Medical records were deidentified and shared with research personnel as described in a previous paper [18]. For the current study, 19 subjects were excluded due to missing data and 11 due to high TT levels at the time of the occupational health evaluation (>916 ng/dL). FFs with high TT were excluded in order to avoid misinterpretation or artificial overestimation of associations among the variables because of the possibility of unreported exogenous testosterone supplementation or over-the-counter testosterone stimulants. Also, 8 subjects were excluded for reporting the use of testosterone formulations (n = 6) or medications that induce testosterone production (clomiphene and anastrazole; n = 2). Another subject using an insulin pump was excluded. Thus, our final sample consisted of 301 male career FFs. 

### 2.2. Measurements and Analysis

Methods detailing the measurement of height, weight, resting blood pressure, and blood work are presented in previous publications [18,19]. Blood samples were obtained from FFs in the morning, after eight to ten hours of fasting, and TT was measured using a standard electrochemiluminescence assay (ECLIA, Roche Diagnostics, Indianapolis, IN, USA) based on the standardization of the assay to the CDC reference method (labcorp.com/assets/11476, accessed on 15 November 2023). All blood analyses were performed by LabCorp (Laboratory Corporation of America, Burlington, NC, USA). Based on the Endocrine Society recommendation [21,22], and in line with our previous study [21], TT values were categorized as: low (<264 ng/dL), borderline (264–399 ng/dL), and reference range (400–916 ng/dL). Given that reference ranges for TT levels in men may vary considerably and different cut-off points have been proposed by various groups [22,23,24], we also performed other comparisons grouping the subjects in the following categories: low + borderline vs. reference range; low TT vs. borderline + reference range; low vs. reference groups; and those with TT < 320 ng/dL as the low group vs. those with TT ≥ 320 ng/dL as the referent group. 

PF tests were performed based on the IAFC/IAFF Wellness Fitness Initiative guidelines. CRF was estimated based on the exercise stage achieved on a Bruce treadmill test using 85% of the age-adjusted maximal heart rate. Due to limitations in analyzing estimated metabolic equivalents based mainly on the test’s stage, we evaluated total exercise time instead. BC was assessed by body fat percentage (BF%). According to standard procedures [25], the chest, abdomen, and thigh skinfolds were measured using a skinfold caliper, and BF% was estimated based on the Jackson and Pollock three-site skinfold formula (Lange, Beta Technology, Santa Cruz, CA, USA). Firefighters with BF% > 25% were considered as having obesity [26,27].

Muscular strength was evaluated based on handgrip dynamometers, and muscular endurance was assessed via push-ups and “plank” tests. In the “plank” exercise, the FF positioned himself by putting his weight on forearms and toes while maintaining his back flat for the maximum time possible and maintaining the adequate technique. The subjects had to maintain a straight line from the head to the toes with an extended leg position, keeping the head and spine in a neutral position. An experienced American College of Sports Medicine certified evaluator monitored the FF’s hip, low, and upper back displacement to ensure alignment during the test. Push-ups were performed according to standard procedures, with subjects on their hand and toes [25]. Volunteers performed the maximal number of repetitions touching a push-up block with their chest and maintaining a proper form and cadence of 80 bpm, until volitional fatigue, or when FFs were unable to maintain the adequate technique. For the handgrip assessment, a hydraulic handgrip dynamometer (Jamar^®^) was adjusted to each firefighter’s hand. Subjects performed a maximal contraction for 2–4 s while holding the handgrip with the arm to the side of their body without touching their thigh and bent at approximately 90° with the gauge facing the evaluator. Subjects performed two attempts with each hand with the highest value being registered. For the analysis, the value of the sum of the two hands was considered, as recommended by the American College of Sports Medicine. For descriptive purposes, we classified the push-up and total handgrip strength results following sex and age normative data [25]. 

### 2.3. Statistical Analyses

The Kolmogorov–Smirnov test was used to confirm the normality of the variables in the groups and subgroups. If a variable was found not to be normally distributed using the Kolmogorov–Smirnov test (*p* < 0.05), it was further examined using the Q-Q Plot [28], and normality was confirmed. Homogeneity of variance was assessed using Levene’s test in all groups and subgroups. Descriptive statistics are presented as continuous and categorical variables. Due to some variables not showing homogeneity of variance, we compared the effect of different TT categories on PF using a robust version of the ANOVA (Welch’s F). We performed a bias-corrected accelerated (BCa) bootstrapping estimation of the 95% confidence interval of the mean difference between variables (post-hoc pairwise comparisons). This analysis was performed if the robust ANOVA showed a significant difference between groups. Since we expected significant differences in age and body composition between TT groups [18,19], we also performed an age- and BF%-adjusted comparison by adjusting for both covariates since they can influence PF. This ANCOVA was followed by a post-hoc pairwise comparison using the same criteria from the ANOVA. Lastly, we performed an exploratory analysis using an independent T-test on different groups using different testosterone cut-offs, as described above.

As another method to control for the effect of body composition on the fitness variables analyzed, we also compared PF between TT groups by BF% categories [normal (BF% ≤ 25.0%) vs. obese (BF% > 25.0%)]. We performed crude and adjusted analyses but this time only using age as a covariate, since we were examining differences within the same BF% categories. Similarly, to control for the effect of age on PF, we compared TT groups by two age categories (≥45 vs. <45 years old) in a crude analysis and an adjusted analysis by BF%. This cut-off point was chosen due to it being a classical cardiovascular disease risk factor [25] and due to the association of age and reduction in testosterone levels [23]. A chi-squared test was used to analyze the association between low fitness level (splitting the groups by the less or equal to the median value in each PF variable) and low TT (<264 ng/dL). The crude and adjusted odds ratios (OR) and 95% confidence intervals (95% CI) were calculated to express the strength of the association. 

The sample size varied between 298 to 301 for different analyses due to missing data in specific variables. The IBM SPSS Statistics^®^ 21 (IBM Corporation, Armonk, NY, USA) software package was used for data processing and analysis and Prism 8 for Windows (GraphPad Software, San Diego, CA, USA) for graphic design. Differences were considered statistically significant when a two-tailed *p*-value was less than 5% (<0.05).

## 3. Results

### 3.1. Descriptive Characteristics

The study population characteristics are presented in Table 1. Participants’ ages ranged from 19 to 60 years old, with the vast majority of FF (99.7%) being less than 60 years old. Despite the relatively low average age of the group (37.5 ± 10.2 years old), participants showed an undesirably high level of CM risk factors (Table 1). Specifically, 21.6% (n = 65) of FFs had obesity, and 35.5% (n = 107) had abnormal resting blood pressure (≥140 mmHg systolic and/or ≥90 mmHg diastolic). Regarding medication, 12.3% (n = 37) reported that they had been treated with anti-hypertensive drugs, 5.0% (n = 15) with lipid-lowering agents, and 1.3% (n = 4) with anti-diabetics. In 1.7% (n = 5) of the participants, we had no data on medication use.

We observed a mean TT value in the reference range but with a large interindividual variation (451.9 ± 161.7 ng/dL) with 32 participants (10.6%) in the low range of TT, 84 (27.9%) in the borderline, and 185 (61.5%) in the reference range of TT. 

Physical fitness variables are shown in Table 2. Considering the mean age of our sample, the average result of the total grip strength and push-ups should correspond to the excellent and very good classification, respectively. The Bruce treadmill test duration time should be equivalent to terminating the Bruce protocol in its third stage at a workload that predicts a CRF of 10 METs. All tests, but especially the plank and push ups tests, had a high degree of variability, as evidenced through their high standard deviation.

### 3.2. Relationship between Physical Fitness and Testosterone

Comparisons of PF variables based on TT category are shown in Table 3. All PF variables, except handgrip strength, were significantly different across TT categories. Plank time and the number of push-ups increased in a stepwise fashion, with those in the reference range achieving a plank time 17 s higher than the low TT group and completing about four more push-ups. The Bruce treadmill test duration was almost 1 min longer (58 s) in the reference range group versus the low TT group. 

Importantly, BF% and age differed in a declining stepwise fashion from the low to the reference range TT groups. After confirming the significant difference in age and BF% between TT groups (*p* < 0.01), we performed another analysis using these variables as covariates. After adjusting for age and BF%, PF variables were no longer statistically significantly different between TT categories. The estimated data from this ANCOVA are presented in Table 4.

To better understand the influence of body composition on physical fitness variables, we also analyzed difference in fitness variables across TT categories within different BF% categories (Table 3: no obesity and obesity groups). In the unadjusted analysis, only the Bruce’s treadmill test duration in the non-obese group was significantly different across TT categories (shorter test time in the low TT group), although both plank time and number of push-ups nearly achieved statistical significance in the non-obese group. In the subsequent analysis that adjusted for age, we observed no significant difference in any PF variables across TT categories, although there was a strong trend of plank time being lower in those with lower TT (Table 4). Notably, there was a large inter-individual subject variation for all PF variables, with the standard deviation being larger than the average difference between groups, limiting the observation of a significant difference between TT categories in both groups—especially considering that the low TT obesity group was limited to 12 FFs.

Similar to what was undertaken for body composition, to better understand the influence of age on PF variables, we also analyzed differences in fitness variables across TT levels within different age categories (<45 years vs. ≥45 years—Table 5). Both in younger and older FFs, the PF test was similar across different groups based on TT values. However, there was a *p*-value of 0.05 for the Bruce treadmill test duration for FFs who were <45 years and a strong trend (*p* = 0.09) for push-ups in FFs ≥ 45 years. After adjusting for body composition, there were no significant differences (or trends) between different TT categories in either the young or older group. The data from this ANCOVA are presented in Table 6.

We also analyzed whether “less fit” individuals were at higher odds of having low TT than the “fitter” FFs considering the entire sample (Figure 1A) or only the low and reference range TT groups (Figure 1B). In both comparisons, FFs with shorter Bruce Treadmill test durations, a proxy for lower CRF, were at significantly higher odds of having low total testosterone (O.R. 3.24–3.41; *p* < 0.01). FFs performing less push-ups were at significantly higher odds of having low total testosterone considering only the reference range and low testosterone groups (O.R. 2.27 [95%CI 1.01–5.08]; *p* < 0.05). The inclusion of individuals with borderline serum total testosterone levels and mixed fitness results might have brought “noise” into the data, since the results were similar but did not reach statistical significance, only a strong trend (O.R. 2.12 [95%CI 0.96–4.67]; *p* = 0.06). There were no significant associations between low total testosterone and total grip strength or plank time (*p* > 0.23). Similar to previous analyses, after adjusting for age and BF%, none remained significant (*p* > 0.15).

Lastly, we investigated whether older FFs, or those classified as obese, had greater odds of having low TT (Figure 2). Both conditions, in both comparisons, were associated with higher odds of having low TT (O.R. 2.41–2.87; *p* < 0.02) in the crude analyses. When the association between obesity and low TT was adjusted for age and the association between age ≥ 45 and low TT was adjusted for BF%, statistical significance was not achieved, although a strong trend remained in the comparison including only FFs in the low testosterone and reference range (*p* > 0.07; Figure 2B).

## 4. Discussion

The main findings of this study are that the low and borderline TT groups had lower PF (push-ups and Bruce Treadmill test duration) compared to the TT reference range group. However, after adjusting for age and BF%, the differences in PF did not achieve statistical significance. Similarly, all significant odds of less-fit FFs having low TT lost significance after adjustment for the same confounders. Therefore, our data suggest that age and body fat may mediate the association between PF and TT. 

TT deficiency and CM diseases share common risk factors such as obesity, age, and metabolic syndrome components [16,21,29,30,31]. These risk factors are also related to PF, although the directionality of the relationship is often unclear [32,33]. Although a direct or reverse causality is yet to be clarified, our data support the hypothesis of a complex and negative vicious cycle among these factors, similar to the one proposed by Genchi et al. According to this model, the relationship between obesity and hypogonadism potentiates a feedback loop so that each condition further exacerbates the other [34]. Specifically, excess body fat is associated with impaired TT levels while hypogonadism, in turn, is associated with fat accumulation, thus leading to a vicious cycle of obesity and hypogonadism [34]. Furthermore, our data support that low PF is associated with obesity and concurrent low TT, thus adding a novel reinforcing factor to the driving of the negative vicious cycle of obesity, low TT, and low PF, with age interacting with all these three components. 

Despite the high physical demand of FFs’ job-related activities and formal recommendations for fitness assessment and minimum standards (NFPA 1582 and 1583) [3,35], the prevalence of obese and unfit FFs is surprisingly high [4,12]. Obesity prevalence in the fire service has been reported from ≈15–37% in Brazil and South Africa and up to 50% in the US [26,36,37,38]. Among US FFs, Storer observed that only 33% achieved the recommended CRF for FFs (CRF ≥ 12 MET), while this proportion was higher but still low (≈50–70%) among Brazilian, Spanish, and Korean FFs [5,6,38]. Also, studies have shown a strong association between PF and body composition (BC) in the general population and FFs [13,38,39]. 

The CM health of FFs has been investigated primarily by assessing traditional cardiovascular (CV) risk factors. Moffatt et al. found that the CV risk profile of a large cohort of US career FFs (4279 males) was similar to that of the general population, which is highly concerning in this workforce [40]. A study aiming to evaluate blood pressure (BP) by decade of life in US career FFs, in comparison with the general population, found a much higher prevalence of hypertension among male FFs compared to the general population in all ages [41]. When analyzing the effect of both BC and age on the overall CM risk factors assessment among US FFs, a study found that a higher BMI, independent of age, was associated with a higher prevalence of CM risk factors and metabolic syndrome [42]. Notably, a recent study among US FFs showed that the group of FFs who lost weight after 5 years of observation (>3% body weight) significantly improved their CM risk profile. At the same time, those who gained weight (>3% body weight) showed increases in total cholesterol, LDL cholesterol, blood glucose, and BMI [43], reinforcing the hypothesis that weight control could be a key CM risk profile mediator. 

Obesity is an inflammatory condition that contributes to atherosclerosis, which is a disease that is responsible for millions of CV deaths per year [44,45]. Besides its direct effects on increasing CM risk, obesity-induced systemic inflammation is also associated with reduced TT levels [34,46]. In a previous study, we observed that the proportion of FFs with low TT that had a normal BMI was only 3.1%, while 59.4% of those with low TT were obese (BMI ≥ 30 km/m^2^) [19]. It is then plausible to consider that, although the link between TT levels and PF might be indirect, the potential of PF mitigating the negative effect of high BF% on testosterone is consistent and represents a positive health parameter. Importantly, despite the great progress of pharmacological alternatives developed in the last decades to treat obesity [47], preventing weight gain must be undoubtedly prioritized. 

Because FFs perform strenuous work that requires muscular and cardiorespiratory fitness, low TT is of great concern in this workforce since testosterone exerts physiological effects on many sites [20], including the CV system, skeletal muscle, and adipose tissue. Symptoms associated with sustained low TT include fatigue, increased body fat, and lowered fat muscle mass [21,22]. The physiological effects of TT, in addition to the fact that PF may improve in men with testosterone deficiency after TRT [48,49], support the general idea that higher TT levels are associated with better PF or vice-versa. In fact, several studies have observed a positive relationship between TT and PF variables, although they often lacked adjustments for age or adiposity [50,51,52]. Others also supported the hypothesis that the relationship between TT levels and key PF variables is mediated by age and body composition. In a study that investigated the relationship between adiposity, CRF, and serum TT levels, bivariate correlations were moderately significant when CRF was controlled for BF%, or vice-versa, but remained weaker for testosterone itself [53]. Similar to our results, this finding also suggests that body fat mediates the association between CRF with serum TT. In a study performed by Koch (2011) with 624 men aged 25 to 85 years old, serum TT was not associated with aerobic performance, which was measured through a progressive incremental exercise protocol on a cycle ergometer [54]. In addition, a study comparing health-related physical fitness components in young and middle-aged men clinically diagnosed with congenital hypogonadotropic hypogonadism (CHH) and clinically healthy controls showed that acute fluctuations in TT levels did not correlate with muscle strength and endurance, neither in men with CHH nor in healthy individuals [55]. Furthermore, an acute intramuscular injection of testosterone esters in healthy eugonadal men did not acutely enhance strength and power [56]. 

In agreement with other studies, our data refuse the hypothesis of a primary association between a low TT measurement with poor PF, but also support the concurrent negative effects of age and body fat on TT levels and PF. These findings challenge the commonsense idea that higher blood concentrations of TT are associated with higher global strength and must be interpreted with caution. Our adjusted data support the previously proposed model of a pathological vicious cycle of obesity, low TT, and low PF with age interacting with all these components. However, in the face of the current research evidence, it is important to note that a consensus regarding the association of testosterone levels and PF is still lacking. A study using the Cooper Center data of 2994 healthy men aged 50–79 years found that men with high CRF (maximal treadmill test) had significantly lower odds (OR: 0.75, 95% CI: 0.71, 0.79) of having low TT (<250 ng/dL) compared to the less fit ones, even after controlling for age, BMI, and current smoking [57]. The higher age of the Cooper Center study (50 to 79 years) and the difference in the CRF assessment may explain the different findings. 

The decline of TT with aging is well accepted in the literature [16,22,58]. Although some uncertainties continue regarding the effects of age on total testosterone concentration [59], low TT in aging men has been shown to be a marker of CV risk [16]. Similarly, BC and PF also tend to worsen with aging [42,60,61]. Although age is a non-modifiable CM risk factor, it is a factor that needs more attention from the Fire Service. A recent study pointed out that fire departments will shortly have more FFs working beyond 60 years due to changes in retirement rules in different countries [33]. This study showed a significant decrease of CRF (−4.42 mL·(min·kg)^−1^; ≈−1.3 MET) and a simultaneous increase in BMI (+1.25 kg/m^2^) after 8 years of follow-up of a relatively young cohort of Brazilian military male FFs (36.2 years) [33]. 

### Limitations

Some limitations in our study must be considered. Firstly, our analyses are from a cross-sectional study design, which precludes causal relationships and the exclusion of reverse causality. Secondly, inherent to the retrospective study design, we used a single testosterone measurement that precludes evidence of an unequivocal state of testosterone deficiency. However, our aim was not to establish a medical diagnosis of low TT but to perform an exploratory analysis of data that were collected with standard procedures in an occupational health setting that is expanding their routine health assessment on FFs. Also, the definition of TT cut-off points is still controversial and might influence the results. To address this inherent limitation, we conducted additional analyses with different TT cut-off points (analyses not shown for simplicity) to determine TT groups, and the results were essentially the same. That is, some PF variables showed poor values in the lowest TT groups in the crude analysis, but no statistical difference remained after adjustment for age and BF%. The assessment of CRF was based on a common practice of only having FF exercise to 85% of the predicted maximum, which may affect the accuracy of the data and affect our understanding of the relationship between CRF and TT. Our comprehensive analyses (recommended and alternative TT cut-off points, subgroups by age and body composition categories) mitigate the effects of these inherent limitations. Although our results are based on data from a convenience sample, which imposes an external validity limitation, our data came from the records of regular medical screening examinations in an entire department that was not subject to any selection bias. Finally, it is important to consider that although the PF tests are validated and widely used among fire service personnel, several of them are affected by BC, making it challenging to evaluate exclusively CRF or muscular endurance. We attempted to overcome such limitations by adjusting analyses for age and BF% and analyzing by body composition and age categories.

## 5. Conclusions

In this cross-sectional study among male career US FFs, we found significant associations between low TT and decreased muscular endurance and cardiorespiratory fitness; i.e., in crude analyses, the low and borderline TT groups had lower PF compared to the TT reference range group. However, after adjusting for age and BF%, no difference remained significant. Therefore, in the final adjusted model, there was no significant association between TT and PF. The adjusted data suggest that body fat and age interact to modulate the significance of testosterone association with a poor PF. In this context, it is important to highlight that excess body fat is responsive to lifestyle modification, which includes a healthy diet and PF improvement [62,63] and represents a priority action to prevent weight gain. Importantly, and considering how difficult it is to treat obesity, even in the context of the progress of pharmacological alternatives developed in the last decades [47,52], preventing weight gain must be prioritized.

Our exploratory and novel findings are relevant to the Fire Service, occupational clinics, and all health professionals who work with firefighters and also from a public health perspective. Our findings support the hypothesis that the association of low TT and impaired PF is mediated largely by age and body fat. Medical and allied health professionals (e.g., medical personnel, exercise physiologists, and dietitians) should be aware of the association between increased adiposity, age, and low testosterone and the potential positive role of reducing body fat and increasing PF as strategies to mitigate the effects of the negative vicious cycle of obesity, low TT, and low PF. Since age is a non-modifiable risk factor, these professionals should target weight loss to combat systemic inflammation, improve endocrine function, and increase physical fitness parameters. 

## Figures and Tables

**Figure 1 ijerph-21-00274-f001:**
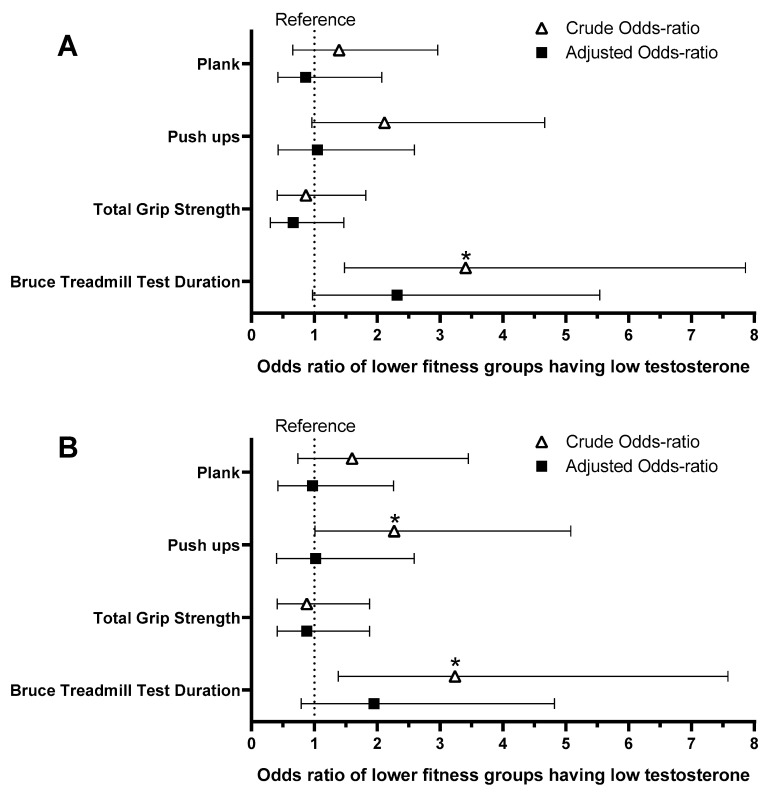
The association between having low fitness values (≤median values) and having low total testosterone (<264 ng/dL). The reference groups for each physical fitness test are composed of individuals who performed better than the median value of the sample; e.g., those with a shorter Bruce treadmill test duration were at higher odds of having low testosterone compared to those with longer test durations in the crude analysis. The adjusted analysis includes age and body fat percentage values as covariates. (**A**): Analysis including the entire sample; (**B**): Analysis including only the low total testosterone and reference range (400–916 ng/dL) groups; *: Significant association with *p* < 0.05.

**Figure 2 ijerph-21-00274-f002:**
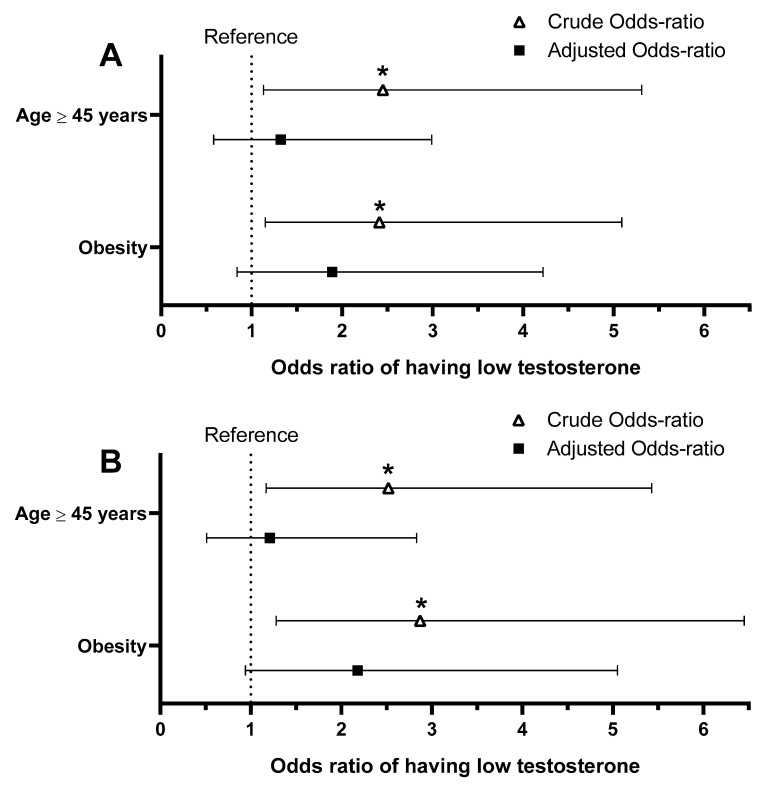
The strength of the association between having one of two different cardiometabolic risk factors (age ≥ 45 years or obesity) and having low total testosterone (<264 ng/dL). For the age risk factor, the reference is the firefighters < 45 years old. For the obesity risk factors, the reference is the firefighters without obesity. The adjusted analysis for the obesity odds ratio included age, while the adjusted analysis for age ≥ 45 years odds ratio used body fat percentage. (**A**): Analysis including the entire sample; (**B**): Analysis including only the low total testosterone and reference range (400–916 ng/dL) groups; *: Significant association with *p* < 0.05.

**Table 1 ijerph-21-00274-t001:** Characteristics of the study population (n = 301).

Continuous Variables	
Age (years)	37.5 ± 10.2
Systolic BP (mmHg)	121.4 ± 7.5
Diastolic BP (mmHg)	77.2 ± 5.5
Body fat percentage (%)	19.3 ± 6.4
Serum total testosterone (ng/dL)	451.9 ± 161.7
**Categorical Variables**	
Age ≥ 45 years old—n (%)	87 (28.9)
Obesity—n (%)	65 (21.6)
TT below reference (<400 ng/dL)—n (%)	116 (38.5)

Continuous variables are expressed as mean ± standard deviation. BP: blood pressure; TT: endogenous testosterone level.

**Table 2 ijerph-21-00274-t002:** Physical fitness characteristics of the study population (n = 301).

Plank ^1^ (s)	86.1 ± 40.5
Push ups ^1^ (repetitions)	26.6 ± 10.5
Total grip strength ^2^ (kg)	119.5 ± 17.7
Bruce Treadmill Test Duration ^2^ (s)	546.8 ± 100.7

Different sample sizes for the variables: ^1^ Sample size is n = 296; ^2^ Sample size is n = 299.

**Table 3 ijerph-21-00274-t003:** Comparison of physical fitness data based on serum total testosterone category for the whole sample and broken down by the presence of obesity.

	Low TT (<264 ng/dL)	Borderline TT (264–399 ng/dL)	Reference Range TT (400–916 ng/dL)	*p*-Value
**Whole Sample (n = 301)**	**n = 32**	**n = 84**	**n = 185**	
Plank ^1^ (s)	72.5 ± 33.3 ^a^	83.9 ± 41.2	89.4 ± 40.9	0.05
Push ups ^1^ (repetitions)	23.2 ± 11.8 ^a^	25.1 ± 8.7 ^a^	27.9 ± 10.8	0.02
Total grip strength ^2^ (kg)	119.7 ± 22.5	120.0 ± 16.1	119.3 ± 17.6	0.95
Bruce Treadmill Test Duration ^3^ (s)	497.9 ± 94.0 ^a,b^	544.6 ± 100.4	556.3 ± 99.9	0.01
**No Obesity (n = 236)**	**n = 20**	**n = 63**	**n = 153**	
Plank ^4^ (s)	73.1 ± 38.6	85.9 ± 38.8	94.1 ± 41.9	0.06
Push ups ^4^ (repetitions)	24.9 ± 12.9	27.0 ± 8.1	29.7 ± 10.4	0.07
Total grip strength ^5^ (kg)	121.0 ± 22.6	120.6 ± 15.9	119.3 ± 17.0	0.83
Bruce Treadmill Test Duration ^6^ (s)	504.0 ± 89.5 ^a,b^	563.8 ± 95.8	572.0 ± 98.4	0.01
**Obesity (n = 65)**	**n = 12**	**n = 21**	**n = 32**	
Plank ^7^ (s)	71.6 ± 22.1	77.7 ± 48.5	67.0 ± 26.9	0.63
Push ups ^7^ (repetitions)	20.0 ± 9.3	18.9 ± 7.9	19.7 ± 9.0	0.93
Total grip strength ^8^ (kg)	117.3 ± 23.0	118.2 ± 16.9	119.5 ± 20.5	0.95
Bruce Treadmill Test Duration (s)	487.8 ± 104.5	488.9 ± 94.5	485.1 ± 73.1	0.99

^a^ Significantly different when compared to the Reference Range group (bias-corrected accelerated bootstrapping post-hoc *p*-value); ^b^ Significantly different when compared to the borderline group; Different sample sizes for the variables: ^1^ Low TT (n = 31), Borderline TT (n = 82), and Reference Range TT (n = 183); ^2^ Low TT (n = 31); ^3^ Borderline TT (n = 82); ^4^ Borderline TT (n = 62) and Reference Range TT (n = 151); ^5^ Reference Range TT (n = 152); ^6^ Borderline TT (n = 61); ^7^ Low TT (n = 11) and Borderline TT (n = 20); ^8^ Low TT (n = 11).

**Table 4 ijerph-21-00274-t004:** Body fat and age-adjusted comparison of physical fitness data based on serum total testosterone category for the whole sample and broken down by the presence of obesity.

	Low TT (< 264 ng/dL)	Borderline TT (264–399 ng/dL)	Reference Range TT (400–916 ng/dL)	Adj. *p*-Value
**Entire Sample (n = 301)**	**n = 32**	**n = 84**	**n = 185**	*
Plank ^1^ (s)	80.5 ± 33.2	88.3 ± 41.0	86.0 ± 37.8	0.62
Push ups ^1^ (repetitions)	26.6 ± 11.9	26.3 ± 8.0	26.8 ± 10.2	0.94
Total grip strength ^2^ (kg)	120.9 ± 22.9	119.8 ± 16.1	119.2 ± 17.6	0.87
Bruce Treadmill Test Duration ^3^ (s)	527.0 ± 94.9	554.3 ± 97.7	546.9 ± 94.9	0.37
**No Obesity (n = 236)**	**n = 20**	**n = 63**	**n = 153**	#
Plank ^4^ (s)	72.3 ± 39.4	85.9 ± 40.0	94.2 ± 42.5	0.06
Push ups ^4^ (repetitions)	26.2 ± 11.9	27.0 ± 8.0	29.5 ± 10.3	0.14
Total grip strength ^5^ (kg)	123.2 ± 22.0	120.7 ± 15.8	118.9 ± 16.9	0.52
Bruce Treadmill Test Duration ^6^ (s)	516.9 ± 92.7	563.7 ± 93.5	569.5 ± 103.2	0.08
**Obesity (n = 65)**	**n = 12**	**n = 21**	**n = 32**	#
Plank ^7^ (s)	71.5 ± 23.2	77.7 ± 45.6	67.1 ± 27.5	0.58
Push ups ^7^ (repetitions)	19.9 ± 8.0	18.8 ± 7.1	19.8 ± 9.1	0.90
Total grip strength ^8^ (kg)	117.0 ± 23.8	118.2 ± 14.4	119.6 ± 20.5	0.92
Bruce Treadmill Test Duration (s)	488.9 ± 99.3	488.4 ± 88.7	485.0 ± 73.9	0.99

*: Body-fat and age-adjusted; #: age-adjusted; Different sample sizes for the variables: ^1^ Low TT (n = 31), Borderline TT (n = 82), and Reference Range TT (n = 183); ^2^ Low TT (n = 31); ^3^ Borderline TT (n = 82); ^4^ Borderline TT (n = 62) and Reference Range TT (n = 151); ^5^ Reference Range TT (n = 152); ^6^ Borderline TT (n = 61); ^7^ Low TT (n = 11) and Borderline TT (n = 20); ^8^ Low TT (n = 11).

**Table 5 ijerph-21-00274-t005:** Comparison of physical fitness data based on serum total testosterone category for the whole sample and broken down by age groups.

	Low TT (<264 ng/dL)	Borderline TT (264–399 ng/dL)	Reference Range TT (400–916 ng/dL)	*p*-Value
**<45 years old (n = 214)**	**n = 17**	**n = 60**	**n = 137**	
Plank ^1^ (s)	72.9 ± 29.2	87.9 ± 43.3	89.1 ± 39.1	0.13
Push ups ^1^ (repetitions)	27.2 ± 10.3	27.1 ± 8.4	29.4 ± 10.8	0.26
Total grip strength ^1^ (kg)	124.3 ± 22.4	123.3 ± 15.9	121.1 ± 16.4	0.62
Bruce Treadmill Test Duration ^2^ (s)	510.3 ± 97.3	566.0 ± 90.8	573.7 ± 97.9	0.05
**≥45 years old (n = 87)**	**n = 15**	**n = 24**	**n = 48**	
Plank ^3^ (s)	72.1 ± 38.7	72.9 ± 33.4	90.2 ± 46.4	0.16
Push ups ^3^ (repetitions)	18.2 ± 12.0	19.6 ± 7.1	23.7 ± 9.6	0.09
Total grip strength ^4^ (kg)	114.0 ± 22.1	111.9 ± 13.7	114.2 ± 19.9	0.84
Bruce Treadmill Test Duration ^5^ (s)	483.9 ± 91.4	489.8 ± 105.2	506.4 ± 89.1	0.64

Different sample sizes for the variables: ^1^ Reference Range TT (n = 136); ^2^ Borderline TT (n = 59); ^3^ Low TT (n = 14), Borderline TT (n = 22), and Reference Range TT (n = 47); ^4^ Low TT (n = 14); ^5^ Borderline TT (n = 23).

**Table 6 ijerph-21-00274-t006:** Body fat-adjusted comparison of physical fitness data based on serum total testosterone category for the whole sample and broken down by age groups.

	Low TT (<264 ng/dL)	Borderline TT (264–399 ng/dL)	Reference Range TT (400–916 ng/dL)	Adj. *p*-Value
**<45 years old (n = 214)**	**n = 17**	**n = 60**	**n = 137**	
Plank (s) ^1^	83.0 ± 30.3	92.2 ± 42.6	85.9 ± 35.9	0.49
Push ups ^1^ (repetitions)	30.4 ± 10.3	28.4 ± 8.0	28.4 ± 10.1	0.70
Total grip strength ^1^ (kg)	123.8 ± 23.0	123.1 ± 16.0	121.2 ± 17.2	0.72
Bruce Treadmill Test Duration ^2^ (s)	532.8 ± 95.1	576.1 ± 90.8	566.5 ± 96.9	0.23
**≥45 years old (n = 87)**	**n = 15**	**n = 24**	**n = 48**	
Plank ^3^ (s)	76.6 ± 41.1	78.5 ± 35.9	86.3 ± 37.6	0.59
Push ups ^3^ (repetitions)	19.3 ± 11.1	20.9 ± 7.1	22.8 ± 8.7	0.34
Total grip strength ^4^ (kg)	113.7 ± 22.5	111.4 ± 14.3	114.5 ± 20.8	0.82
Bruce Treadmill Test Duration ^5^ (s)	493.0 ± 74.3	497.8 ± 99.3	499.8 ± 70.6	0.97

Different sample sizes for the variables: ^1^ Reference Range TT (n = 136); ^2^ Borderline TT (n = 59); ^3^ Low TT (n = 14), Borderline TT (n = 22), and Reference Range TT (n = 47); ^4^ Low TT (n = 14); ^5^ Borderline TT (n = 23).

## Data Availability

Data obtained from LifeScan and are available from author with the permission of LifeScan.

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
