# Peer review of "Association of Endogenous Testosterone with Physical Fitness Measures during Firefighter Occupational Health Evaluations"

_ijerph, 2024, doi:10.3390/ijerph21030274_

Round 1
Reviewer 1 Report
Comments and Suggestions for Authors
First of all, thank you for the opportunity to review your work. Regardless of the final result of this process (approval or not), the aim is to improve the quality of the manuscript for readers.
Congratulations on your honesty in explaining that the data continues studies 20 and 21. It is possible to extract different research problems with data collected from the same sample.
Your work manuscript can be approved for publication in this journal if it meets the requests. In particular, the rationale in the introduction is based on cardiometabolic parameters and leaves aside strength tests, which are the majority of the study variables.
INTRODUCTION
Lines 46-47 - What is the reason or evidence that the NFPA presents the 12 METS cutoff line? Wouldn't a firefighter with 11 METs be fit? The recommendation >10METs is already considered adequate for performing strenuous activities. Please understand that this is a constructive reflection. Many cutoff lines about physical fitness are presented by military or tactical personnel institutions are arbitrary and without scientific explanation. Please discuss why 12 METs.
Lines 65-66- About: “In the context of the complex CM risk assessment, endogenous serum testosterone (TT) levels have recently appeared as an emerging cardiovascular risk factor.”
TT level as a recent cardiovascular risk factor? Where's the quote? If they are references 18 and 19, they are from 2018. Why would it be something emerging? TT appeared here with a leap of logic. There needs to be more connection and introduction to address this variable.
Line 70 - There is data to produce this presented manuscript. Still, the introduction does not point out the gap in the topic and the objective, which is to measure the association between testosterone concentration and the level of physical fitness. 249 to 836 ng/dL are within normal TT concentration levels. In study 20, the low (<264), borderline (264-399 ng/dL), and reference value (400-916 ng/dL) groupings were stratified as they were associated with ventricular thickness. This introduction needs to explain why these TT concentrations would be associated with handgrip dynamometer/push-ups and “plank” tests. These physical tests do not have cardiovascular characteristics as performance determinants. Lean mass is positively associated with handgrip (PMID: 35442358).
Lines-77-78 Establishing a logical line of the possible association of testosterone concentration with strength tests (handgrip, push-up, and plank) is necessary. In turn, the hypothesis that, due to the association of low testosterone levels, it is associated with a smaller ventricular thickness and that this can compromise aerobic performance (Bruce treadmill test) is implicit throughout the reading.
MATERIALS AND METHODS
Lines 124-127- Replace “plank exercise” with front plank test. About “…with adequate technique.”. Very embracing. Please be more descriptive in the test criteria.
Lines 130-133 - Describe: 1) the dynamometer model. 2) Duration of muscle action. 4) Why did you present the sum and not the unilateral force of one or both hands separately?
Line 137- About “If a variable was suspected not to be normally distributed, it…” The normality test used is dichotomous and depends on the p-value. Please explain further what the criteria for suspicion were. The Kolmogorov-Smirnov confirmed normality, and you suspected it was not? Kolmogorov-Smirnov confirmed there was no normality, and you suspected it was normal?
Line 153 – About ..” TT groups by BF% categories [normal (BF% 153 ≤ 25.0%) vs obese (BF% > 25.0%)].” Where is the description, rationale, or citation for this cutoff point?
RESULTS
Table 1- About: “Age ≥ 45 years old – n (%)”. Where is the description, rationale, or citation for this cutoff point?
Table 1- About: “Obesity – n (%)” Please, Enter the value body fat percentage >25%.
Lines 183-184 – Please, explain the values to obtain this classification in materials and methods. Is it excellent or very good based on what values?
Lines 186-187 – About “METs. All tests, but 186 especially the plank and push ups tests, had a high degree of variability, as evidenced through their high standard deviation.” If the push-ups presented a high variability, they cannot be classified as excellent or very good, correct? (lines 183-184). Please explain.
Lines 197-198 – About: “Test duration was almost one minute longer in the reference group versus the low TT group.” Which test? The plank had a 17s difference between the groups. About: "almost one minute" is not a scientific language for presenting results. This interpretation can be placed in the discussion section.
Lines 207-208 – About: “… only test duration in the non-obese …” Does the test duration refer to the Bruce Treadmill or front plank tests? Please be accurate in your writing. You mixed that sentence with the push-up test (several repetitions).
Table 3- I will use Table 3 as an example, but this needs to be reviewed in all tables. Placing the numbers in the variables and the explanation in the caption was an excellent strategy to explain the number of firefighters who carried out each test. However, it raised a doubt. The Whole Sample presents n = 301. However, when adding No Obesity (n=232) + Obesity (n=65), we have n = 297. Where are the four firefighters from the entire sample? This gets complicated when we check for each variable as well. Please review and explain.
Table 3 – About: “Total grip strength2 (-)”How is it possible that the handgrip test value does not have a unit?
Table 3 – About “Exercise Test Duration3 (s)” . I suggest it be called the Bruce Treadmill test throughout the manuscript. The “Exercise Test Duration” is unclear.
Table 3 – About: “/487.8 ± 104.5” Please remove the bar.
Lines 232-233 – About “… across TT levels 232 within different age categories (< 45 years vs ≥ 45 years - Table 5)…” Did I question the rationale for this age cutoff in Table 1?
DISCUSSION
Lines 392-393 - About “To address this inherent limitation, we conducted additional analyses with different TT cut-off points (data not shown for simplicity) to determine TT groups and the results were essentially the same.” I suggest cluster analysis to separate groups or cutoff points in future studies.
REFERENCE
About: “20. Lofrano-Porto A, Soares EM, Mattias A, Grossi Porto GP, Smith D. Borderline-low testosterone levels are associated with 493 lower left ventricular wall thickness in firefighters: an exploratory analysis. UNDER REVIEW. 2020 [Internet]. [date un-494 known];” No longer in UNDER REVIEW (PMID: 32633472). Correct.
Author Response
Department of Health and Human Physiological Sciences
February 15, 2024
Prof. Dr. Paul B. Tchounwou, PhD
Editor-in-Chief
International Journal of Environmental Research and Public Health – IJERPH
Dear Dr. Tchounwou:
We thank the Editors of IJERPH and the Reviewers for your interest in our manuscript (ASSOCIATION OF ENDOGENOUS TESTOSTERONE WITH PHYSICAL FITNESS MEASURES DURING FIREFIGHTER OCCUPATIONAL HEALTH EVALUATIONS) and for the favorable comments. We appreciate the Reviewers' suggestions and have provided a revised manuscript in response to these helpful comments, with highlighted edits.
We have produced a revised manuscript attempting a balance between both Reviewers' comments. Thus, we addressed all comments and suggestions from Reviewer 1 but also tried to change the text as little as possible, considering the very positive review from Reviewer 2, who made only "…minor, mostly editorial, comments…" in the manuscript. We believe that the response and adjustments in response to the Reviewers' comments and suggestions have resulted in an improved manuscript, which we hope meets the high scientific standard of IJERPH.
Regarding the preliminary issue raised by the Managing Editor related to a "high
proportion of the cited references belong to you or your co-authors," we agree that this is not typically warranted. However, in this instance, most of the authors have had the main topics of this paper (firefighters' cardiometabolic health and/or the role of testosterone on health and diseases) as their main research interest for a long time, and they have published extensively in this area. In fact, the research team was assembled because of these overlapping areas of
expertise. However, considering your legitimate concern we have carefully reviewed the cited references and excluded five of them, as the rationale was supported by other references. We feel that the remaining references should be retained because they support specific points and removing them would require significant modification to our logical arguments and the text.
Specific responses and edits of the manuscript based on reviewer comments are provided below.
Reviewer #1
|
INTRO |
|
|
Lines 46-47 - What is the reason or evidence that the NFPA presents the 12 METS cutoff line? Wouldn't a firefighter with 11METs be fit? The recommendation >10METs is already considered adequate for performing strenuous activities. Please understand that this is a constructive reflection. Many cutoff lines about physical fitness are presented by military or tactical personnel institutions are arbitrary and without scientific explanation. Please discuss why 12 METs. |
We thank the Reviewer for this constructive comment. There is a substantial evidence supporting high energy demands of firefighting. For example, one paper shows the oxygen cost of various firefighting activities and observed values between 8-14 METs (doi: 10.1016/j.apergo.2009.07.009). The 12 MET recommendation is not our recommendation, but we note that it has been adopted by NFPA, which is a worldwide recognized institution related to firefighters' health and an institution that establishes health and performance standards for US firefighters, which are exactly our population. To better clarify this, we added the following text:
“The US National Fire Protection Association (NFPA) recommends a minimum CRF of 12.0 metabolic equivalents (MET) for FFs to safely and efficiently perform their job-related tasks (4); also, this cut-off point has been widely used in firefighter fitness-related research (4–6).”
The 12 MET recommendation has been widely used in research, as shown by the papers mentioned in our discussion (lines 321-325 original manuscript). |
|
TT level as a recent cardiovascular risk factor? Where's the quote? If they are references 18 and 19, they are from 2018. Why would it be something emerging? TT appeared here with a leap of logic. There needs to be more connection and introduction to address this variable. |
We thank the Reviewer also for this comment. The statement that TT level has been investigated as an emerging cardiovascular risk factor is indeed supported by references 16 and 17 and others. We understand the Reviewer´s concern about the year of the references (2018 – i.e., 5 to 6 years old). However, establishing a potential new risk factor takes time. We believe that TT level as an emerging cardiometabolic risk factor is appropriate. Also, we believe that the connection between TT level, firefighters´ health, and physical fitness is appropriately presented in the introduction (lines 45-46 and 62-64 are good examples). More detailed information is presented in the discussion. Again, the decision not to expand the introduction is based on the necessity to balance both Reviewers' comments. |
|
There is data to produce this presented manuscript. Still, the introduction does not point out the gap in the topic and the objective, which is to measure the association between testosterone concentration and the level of physical fitness. 249 to 836 ng/dL are within normal TT concentration levels. In study 20, the low (<264), borderline (264-399 ng/dL), and reference value (400-916 ng/dL) groupings were stratified as they were associated with ventricular thickness. This introduction needs to explain why these TT concentrations would be associated with handgrip dynamometer/push-ups and “plank” tests. These physical tests do not have cardiovascular characteristics as performance determinants. Lean mass is positively associated with handgrip (PMID: 35442358). |
To briefly describe our rationale which is presented in the introduction: • firefighters’ have high job demands that require adequate physical fitness, • firefighters have high cardiometabolic risk, • there is evidence of a potential role of TT on cardiometabolic health and its association with physical performance, • there is a gap in understanding the potential association of TT and physical fitness in a population with high cardiometabolic risk and high physical demands.
Regarding the stratification of the TT groups. This was based on guidelines from major endocrine organizations and is detailed in the methods section (lines 108 – 115). Recognizing that cut-off points can vary, we also included in the manuscript that other cut-off points were tested, and we found the same results. We agree that handgrip dynamometer/push-ups and “plank” tests do not assess cardiovascular performance. But, TT may affect both CRF and muscular fitness. Both CRF and muscular fitness measures are included in our objectives and addressed in the discussion. |
|
Establishing a logical line of the possible association of testosterone concentration with strength tests (handgrip, push-up, and plank) is necessary. In turn, the hypothesis that due to the association of low testosterone levels, it is associated with a smaller ventricular thickness and that this can compromise aerobic performance (Bruce treadmill test) is implicit throughout the reading. |
We thank the Reviewer for this point. We believe that part of it was answered in the previous comment. However, to make the potential association with muscular fitness clearer, we included a new statement in the introduction, as follows: “However, despite the known physiological effects of TT on the cardiovascular and muscular systems (20), whether low TT levels affects PF remains unclear, even in the general population.” |
|
METHODS |
|
|
Lines 124-127- Replace “plank exercise” with front plank test. About “…with adequate technique.”. Very embracing. Please be more descriptive in the test criteria. |
We thank the reviewer for this comment. We edited that section and included the following information: “In the “plank” exercise, the FF would position himself putting his weight on forearms and toes while maintaining the back flat for the maximum time possible maintaining adequate technique. The subjects had to maintain a straight line from the head to the toes with an extended leg position, keeping the head and spine in a neutral position. An experienced American College of Sports Medicine certified evaluator monitored the FF’s hip, low, and upper back displacement to ensure alignment during the test.” |
|
Lines 130-133 - Describe: 1) the dynamometer model. 2)Duration of muscle action. 4) Why did you present the sum and not the unilateral force of one or both hands separately? |
We thank the Reviewer for this comment. We have updated our text, to include the information requested Push-ups were performed according to standard procedures, with subjects on their hand and toes (25). Volunteers performed the maximal number of repetitions touching a push-up block with their chest and maintaining a proper form and cadence of 80 bpm, until volitional fatigue, or when FFs were unable to maintain adequate technique. For the handgrip assessment, a hydraulic handgrip dynamometer (Jamar®) was adjusted to the firefighter’s hand. Subjects performed a maximal contraction for 2-4 seconds while holding the handgrip with the arm to the side of their body without touching their thigh, and bent at approximately 90° with gauge facing the evaluator. Subjects performed two attempts with each hand with the highest value being registered. For the analysis, the value of the sum of the two hands was considered, as recommended by the American College of Sports Medicine. Using age- and sex-specific normative data, we classified the push-up and total handgrip strength test results for descriptive purposes (25). |
|
Line 137- About “If a variable was suspected not to be normally distributed, it…” The normality test used is dichotomous and depends on the p-value. Please explain further what the criteria for suspicion were. The Kolmogorov-Smirnov confirmed, and you suspected it was not? Kolmogorov- Smirnov confirmed there was no normality, and you suspected it was normal? |
We thank the Reviewer for this important comment. We agree that the word “suspected” does not sound appropriate. Our initial hypothesis was that all physical fitness variables would have a normal distribution. Unfortunately, false positives in Kolmogorov-Smirnov (K-S) tests are common in large sample sizes. Thus, in the case of a positive K-S test (p<0.05), we performed the plot analysis to assess normality (which is considered superior to the K-S test). See the updated version: “If a variable was found not to be normally distributed using the Kolmogorov-Smirnov test (p<0.05), it was further examined using the Q-Q Plot (28).” |
|
Line 153 – About ..” TT groups by BF% categories [normal (BF%153 ≤ 25.0%) vs obese (BF% > 25.0%)].” Where is the description, rationale, or citation for this cutoff point? |
We thank the Reviewer for this comment. The use of the BF%>25 cutoff point in men for obesity assessment has been widely used in research. This BF% cutoff point has been used as a gold standard compared to body mass index (BMI) to assess obesity. To illustrate this, we share three different papers on firefighters using the same cutoff point (Poston et al., 2011 [PMID: 21386691]; Porto et al., 2016 [PMID: 27901177]; Gurevich et al., 2017 [PMID: 27694377].) This cut-off point was also used in a meta-analysis to evaluate BMI’s diagnostic performance of obesity. We have now included this last article in our paper and have included the following statement: “According to standard procedures (25), the chest, abdomen, and thigh skinfolds were measured using a skinfold caliper and BF% was estimated based on Jackson and Pollock three site skinfold formula (Lange, Beta Technology, Santa Cruz, CA, USA). Firefighters with BF%> 25% were considered as having obesity (26,27).” |
|
RESULTS |
|
|
Table 1- About: “Age ≥ 45 years old – n (%)”. Where is the description, rationale, or citation for this cutoff point? |
We thank the Reviewer for the insightful comment. We added the following text to the methods: “Similarly, to control for the effect of age on PF, we compared the TT groups by two age categories (≥ 45 vs < 45 years old) in a crude analysis and an adjusted analysis by BF%. This cut-off point was chosen due to being a classical cardiovascular disease risk factor (25) and due to the association of age and reduction in testosterone levels (23).” |
|
Table 1- About: “Obesity – n (%)” Please, Enter the value bodyfat percentage >25%. |
We thank the Reviewer for the suggestion. Despite knowing that there is no universal best cut-off point for BF%-based obesity, we have included a statement explicitly defining our obesity criteria in the methods section; thus, we believe it is unnecessary to make this change. The updated version of the methods section, addressing your suggestion, is now sufficient: Firefighters with BF%> 25% were considered as having obesity (26, 27). |
|
Lines 183-184 – Please, explain the values to obtain this classification in materials and methods. Is it excellent or very good based on what values? |
We thank the Reviewer for this comment. We included the following statement: “For descriptive purposes, we classified the push-up and total handgrip strength results following sex and age normative data (25).” |
|
Lines 186-187 – About “METs. All tests, but 186 especially the plank and push ups tests, had a high degree of variability, as evidenced through their high standard deviation.” If the push-ups presented a high variability, they cannot be classified as excellent or very good, correct? (lines 183-184). Please explain. |
We thank the Reviewer for the comment. To clarify this, we updated the statement as follows: Considering the mean age of our sample, the average result of the total grip strength and push-ups would correspond to the excellent and very good classification, respectively. |
|
Lines 197-198 – About: “Test duration was almost one minute longer in the reference group versus the low TT group.” Which test? The plank had a 17s difference between the groups. About: "Almost one minute" is not a scientific language for presenting results. This interpretation can be placed in the discussion section. |
We thank the Reviewer for the comment. We believe that addressing the terminology of the Bruce treadmill test will solve the issue, making clear that we were referring to the Bruce test. To address both concerns, we modified the sentence to read: Treadmill test duration was almost one minute longer (58 s) in the reference range group versus the low TT group |
|
Lines 207-208 – About: “… only test duration in the non-obese…” Does the test duration refer to the Bruce Treadmill or front plank tests? Please be accurate in your writing. You mixed that sentence with the push-up test (several repetitions). |
We thank the Reviewer for this comment. To facilitate understanding, we made clear in Methods that every time a plank test is being referred to, we mention plank time. Also, for the treadmill test, every time it is referred to, we now use the term “Bruce treadmill test duration.” Changes were made throughout the entire document. |
|
Table 3- I will use Table 3 as an example, but this needs to be reviewed in all tables. Placing the numbers in the variables and the explanation in the caption was an excellent strategy to explain the number of firefighters who carried out each test. However, it raised a doubt. The Whole Sample presents n = 301. However, when adding No Obesity (n=232) + Obesity (n=65), we have n = 297. Where are the four firefighters from the entire sample? This gets complicated when we check for each variable as well. Please review and explain. |
We thank the Reviewer for addressing this. Table 4 correctly stated the No Obesity as (n=236). The same number can be seen in the No Obesity columns of Table 3 (20 + 63 + 153). Table 3 has been updated, and we appreciate the attentiveness to our data. |
|
Table 3 – About: “Total grip strength (- )”How is it possible that the handgrip test value does not have a unit? |
We thank the Reviewer for noticing this. We had included the unit for Table 2, but unfortunately, it was missing for the other tables. They have all been updated. |
|
Table 3 – About “Exercise Test Duration3 (s)” . I suggest it be called the Bruce Treadmill test throughout the manuscript. The “Exercise Test Duration” is unclear. |
We thank the Reviewer for the suggestion. All tables have been updated. |
|
Table 3 – About: “/487.8 ± 104.5” Please remove the bar. |
We thank the Reviewer for noticing this. The text has been updated. |
|
Lines 232-233 – About “… across TT levels 232 within different age categories (< 45 years vs ≥ 45 years - Table 5)…” Did I question the rationale for this age cutoff in Table 1? |
Yes, this issue was pointed out in a previous comment, and we have already addressed it. |
|
DISCUSSION |
|
|
Lines 392-393 - About “To address this inherent limitation, we conducted additional analyses with different TT cut-off points (data not shown for simplicity) to determine TT groups and the results were essentially the same.” I suggest cluster analysis to separate groups or cutoff points in future studies. |
We thank the Reviewer for this suggestion and will keep it in mind for future studies. |
|
REFERENCE |
|
|
About: “ 20. Lofrano-Porto A, Soares EM, Mattias A, Grossi PortoGP, Smith D. Borderline-low testosterone levels are associated with 493 lower left ventricular wall thickness in firefighters: an exploratory analysis. UNDER REVIEW. 2020 [Internet]. [date un-494 known];” No longer in UNDER REVIEW (PMID: 32633472).Correct. |
We thank the Reviewer once again for its attentiveness. The mistake is now corrected. Reference 18 now. |
Reviewer #2
|
Abstract, pg. 1: If the word count allows, you may briefly state the statistical method used for the main associations (ANOVA). Also, would 'confound' be a better choice than 'mediate' in line 30? |
We thank the Reviewer for the suggestion. We are slightly above the current word count with our current 210 words (instructions state about 200 words). Also, we feel that the use of both ANOVA and ANCOVA are important to the statistical analysis, and the addition of both instead of one would be more reflective of the statistical methods employed. But, due to the limits in word count, we feel that despite the excellent suggestion, we prioritized conciseness in this part of the abstract. Following Reviewer´s recommendation, we changed from “mediate” to “confound”. |
|
Page 5: It would be helpful to merge the contents of Tables 1 and 2. |
Again, we thank the Reviewer for this suggestion. However, we believe that keeping both tables separate allows us to show our results distinctly from the participant´s characteristics. Also, our main purpose in keeping both separate is to |
|
facilitate the understanding of further results (Table 3 onward) and how the subgroups distinguish themselves from the larger “population data” represented in our sample |
|
|
The highest testosterone level is considered the ‘referent’ group. Recommend replacing ‘reference’ with ‘referent’ in the Tables and text of the Results. |
Thank you for the suggestion. Changes were made accordingly. The reference group was described as a reference range and was consistent with the “names” of the other groups that were described according to their serum testosterone levels (e.g., borderline low). Thus, to remain consistent with the suggestion, whenever the text mentioned serum testosterone level reference ranges or referred to firefighters with serum total testosterone within the reference range, the term reference was kept or included. Whenever a group was mentioned as a referent of an analysis, it was changed to the referent. |
|
In the Conclusions, the authors stated that “Our findings support the hypothesis that the association of low TT and impaired PF is mediated largely by age and body fat.” It is important that this hypothesis be stated clearly early on, in the last paragraph of the Introduction, after stating the purpose of the study. Also, the authors should make clear in the Abstract, Results, and Conclusion, that the association between testosterone levels and physical fitness was not statistically significant. The final reported results should be the adjusted results; in these models, age and body fat percent are important confounding variables and after inclusion, the association was no longer significant. |
Once more, we thank the Reviewer for highlighting this important point. Even though the abstract is slightly above the recommended 200 words, we agreed with the Reviewer and included a new statement in the conclusion of the abstract and the conclusion of the manuscript to highlight the main finding. Also, we decided to not include this hypothesis in the introduction since it was not an “a priori” hypothesis. |
Thank you very much in advance for your consideration.
Sincerely,
Denise L. Smith, PhD
Health and Human Physiological Sciences
Director First Responder Health and Safety Lab
Skidmore College
phone: +1 (518) 496-7307
e-mail: dsmith@skidmore.edu
Reviewer 2 Report
Comments and Suggestions for Authors
The authors did an excellent job in all aspects of this research. I usually have several comments for authors, however, this is one of the best manuscripts that I’ve agreed to review.
The authors conducted a cross-sectional study to investigate the association between serum testosterone levels and physical fitness among US firefighters. The authors have conducted an excellent study. The paper is well organized, clear details are provided, the statistical methods are accurate, and the results are clearly presented and discussed.
The main objective of the research was to determine the cross-sectional relationship between serum testosterone levels (low, borderline, and high) and physical fitness (e.g., cardiorespiratory fitness, muscular strength, and muscular endurance) among 301 US firefighters. Although previous investigators have assessed associations between testosterone levels and various physical fitness measures, this paper addresses this important topic in an occupational cohort that is exposed to heavy physical stresses.
The research methodology is adequate and is clearly described in the paper. References are relevant, up-to-date, and adequate for the topic. Information is properly cited. The authors stated the implications of obesity levels in the Discussion which, at first, may seem excessive but is directly related to physical fitness, so it’s appropriate. The authors appropriately discussed other studies and how those results supported or contradicted the findings of the current paper. The limitations of the study are discussed in detail.
I have a few minor, mostly editorial, comments for the authors.
Abstract, pg. 1: If the word count allows, you may briefly state the statistical method used for the main associations (ANOVA). Also, would 'confound' be a better choice than 'mediate' in line 30?
Page 5: It would be helpful to merge the contents of Tables 1 and 2.
Introduction, pg. 2, line 75: Recommend replacing ‘influence’ with ‘is associated with’.
The highest testosterone level is considered the ‘referent’ group. Recommend replacing ‘reference’ with ‘referent’ in the Tables and text of the Results.
In the Conclusions, the authors stated that “Our findings support the hypothesis that the association of low TT and impaired PF is mediated largely by age and body fat.” It is important that this hypothesis be stated clearly early on, in the last paragraph of the Introduction, after stating the purpose of the study. Also, the authors should make clear in the Abstract, Results, and Conclusion, that the association between testosterone levels and physical fitness was not statistically significant. The final reported results should be the adjusted results; in these models, age and body fat percent are important confounding variables and after inclusion, the association was no longer significant.
Author Response

(The authors gave the same response as above.)
